# Investigation of the Molecular Mechanisms Underlying the Antiatherogenic Actions of Kaempferol in Human THP-1 Macrophages

**DOI:** 10.3390/ijms23137461

**Published:** 2022-07-05

**Authors:** Etimad Huwait, Maha Ayoub, Sajjad Karim

**Affiliations:** 1Biochemistry Department, Faculty of Sciences, King Abdulaziz University, Jeddah 21589, Saudi Arabia; ehuwait@kau.edu.sa (E.H.); mayoub0004@stu.kau.edu.sa (M.A.); 2Cell Culture Unit and Experimental Biochemistry Unit, King Fahd Medical Research Centre, King Abdulaziz University, Jeddah 21589, Saudi Arabia; 3Center of Excellence in Genomic Medicine Research, King Abdulaziz University, Jeddah 21589, Saudi Arabia; 4Department of Medical Laboratory Technology, Faculty of Applied Medical Sciences, King Abdulaziz University, Jeddah 21589, Saudi Arabia

**Keywords:** kaempferol, human THP-1 macrophages, cardiovascular disease atherosclerosis, Affymetrix microarrays, IFN-γ, MCP-1, ICAM-1

## Abstract

Cardiovascular disease (CVD) is causing high mortality worldwide (World Health Organization-WHO, 2015). Atherosclerosis, the hardening and narrowing of arteries caused by the accumulation of fatty acids and lipids (cholesterol plaques), is a main reason of stroke, myocardial infarction, and angina. Present therapies for cardiovascular disease basically use statins such as β-Hydroxy β-methylglutaryl-CoA, with <70% efficacy and multiple side effects. An in vitro investigation was conducted to evaluate the impact of kaempferol, a natural medication, in an atherosclerotic cell model. We used cytotoxicity assays, Boyden chamber invasion assays, and quantitative PCR. Affymetrix microarrays were used to profile the entire transcriptome of kaempferol-treated cell lines, and Partek Genomic Suite was used to interpret the results. Kaempferol was not cytotoxic to THP-1 macrophages. In comparison to the control, kaempferol reduced monocyte migration mediated by monocyte chemotactic protein 1 (MCP-1) by 80%. The qPCR results showed a 73.7-fold reduction in MCP-1 and a 2.5-fold reduction in intercellular adhesion molecule 1 (ICAM-1) expression in kaempferol-treated cells. In interferon gamma (IFN-γ) without kaempferol and IFN-γ with kaempferol treated cells, we found 295 and 168 differentially expressed genes (DEGs), respectively. According to DEG pathway analysis, kaempferol exhibits anti-atherosclerosis and anti-inflammatory characteristics. Kaempferol is an effective and safe therapy for atherosclerosis.

## 1. Introduction

Atherosclerosis, a slow-progressing disorder, is caused by the accumulation of fatty acids and lipids in the arteries. It is primarily responsible for myocardial infarction, stroke and heart failure [1]. The trapping of low-density lipoprotein (LDL) in the intima is the first inducer of atherosclerosis, accompanied by abundantly expressed ICAM-1 in the epithelium, which aids the binding of circulating monocytes to the accumulated oxidized LDL (oxLDL) [2]. Active endothelial cells also release MCP-1, a pro-inflammatory cytokine that attracts circulating monocytes to the site of accumulated oxLDL [2]. Once these monocytes reach the intima, they specialize into macrophages and are capable of oxLDL absorption, resulting in the production of foam cells, and progressively buildup there. Thrombosis, myocardial infraction (MI), and stroke may result from the formation of unstable plaques, which rupture suddenly [3].

Macrophages induce atherogenesis and perform unique functions. Monocytes penetrate the plaques during atherogenesis and evolve into M1 macrophages with the help of IFN-γ [4]. Due to their low cholesterol efflux capabilities, these cells generate foam cells after ingesting lipoprotein particles. Inflammatory mediators, oxygen radicals, and matrix metalloproteinases produced by foam cells destabilize the atherosclerotic plaques. Atherosclerotic lesion progression and rupture are mediated by subsequent immune cell penetration and smooth muscle cell (SMC) stimulation of foamy macrophages [5]. Dying macrophages promote necrotic core growth and plaque thickness in advanced atherosclerotic lesions. Macrophages that initially enter the plaques can have an opposite phenotype (M2) and are less likely to produce foam cells than M1 macrophages. M2 macrophages are phagocytic and secrete IL-10, an anti-inflammatory cytokine [6]. M2 macrophages, on the other hand, transition to an M1 phenotype when the plaque grows, secreting IL-1, IL-6, and TNF-α as the illness progresses [7]. Polyphenols appear to increase macrophage polarization toward the M2 phenotype, according to growing literature evidence [8,9].

β-Hydroxy-methylglutaryl-CoA (HMG-CoA) reductase is a rate-limiting enzyme in cholesterol production, and statins competitively inhibit it [10]. According to estimates, current statin-based medications are only effective at reducing LDL levels and preventing up to 70% of cardiovascular disease-related events [2,11,12]. However, statin use has negative consequences, such as liver failure, muscle diseases, type 2 diabetes, and so on [13], which emphasizes the need for innovative ways to prevent atherosclerosis.

Natural plant-based solutions have become a new choice for preventing atherosclerosis and improving human health. In the recent past, bioactive substances and their effects on promoting good health have been intensively studied [14,15,16,17]. Kaempferol has been proven in trials to have anti-atherosclerotic and anti-inflammatory properties [18,19]. Kaempferol is protective for blood vessels against damages caused by oxidative stress and inflammation [20]. We investigated cell survival, cell proliferation, cell migration, cholesterol efflux, and the production of pro-inflammatory mediators of ICAM-1 and MCP-1 to determine the anti-inflammatory or anti-hyperlipidemic effects of kaempferol on THP-1 macrophages.

## 2. Results

### 2.1. Influence of Kaempferol on the Survival of THP-1 Macrophages

A crystal violet test and an LDH assay were used to examine the viability of monocyte-derived THP-1 macrophages. No significant changes were found in LDH levels after exposure to various doses of kaempferol (Figure 1a). However, the release and enzymatic activity of intracellular LDH, a stable cytoplasmic enzyme, were examined to see if varied dosages of kaempferol caused cytotoxicity in THP-1 cells. No cytotoxicity was observed compared to vehicle-treated cells. Because crystal violet can only bind to the DNA of living cells, the crystal violet assay was employed to measure cell survival of treated THP-1 at various dosages of kaempferol. In comparison to vehicle-treated cells, kaempferol-treated cells revealed no significant variations in cell proliferation (Figure 1b).

### 2.2. Downregulation of MCP-1 and ICAM-1 Expression by Kaempferol

IFN-γ stimulated THP-1 macrophages, which were then treated with 5 and 10 µM kaempferol or a vehicle control (0.1% DMSO). The results showed that activating THP-1 cells with IFN-γ alone increased MCP-1 expression by 73.71-fold and ICAM-1 expression by 2.47-fold. Pretreatment of THP-1 macrophages with IFN-γ before adding kaempferol resulted in a 2.56-fold (*p* < 0.0001) and 2.80-fold (*p* < 0.0001) reduction in ICAM-1 compared to treatment with IFN-γ alone (Figure 2a). Furthermore, compared to IFN-γ treatment, kaempferol treatment at the two chosen concentrations resulted in a 70.41-fold (*p* < 0.0001) and 72.19-fold (*p* < 0.0001) drop in MCP-1 (Figure 2b).

### 2.3. Inhibition of the Transferring of Monocytes by Kaempferol

A migration experiment demonstrated a 53% (*p* < 0.0001) decrease in the number of invading cells after kaempferol administration (Figure 3). The addition of MCP-1, on the other hand, resulted in a significant increase in monocyte migration of up to 80% (*p* < 0.0001) as compared to the control sample.

### 2.4. Differential Gene Expression Profiles in Differentiated THP-1 Are Induced by Kaempferol

We considered four conditions (Q1, Q2, Q3, Q4) for expression profiling against three vehicle controls (C1, C2, C3) treated with DMSO. Q1 and Q2 samples were treated with IFN-γ only, while Q3 and Q4 samples were treated with IFN-γ and kaempferol. Assuming the criterion of *p*-value of 0.05 and fold change > ±2, we discovered 295 differentially expressed genes (DEGs), of which 266 were upregulated and 29 were downregulated by IFN-γ treatment, and 168 DEGs, of which 113 upregulated and 55 downregulated by IFN-γ + kaempferol treatment (Table 1 and Table 2). The number of DEGs was reduced to 256 and 156 after removing duplicates, and the overlapping of two list of DEGs from IFN-γ treatment and IFN-γ + kaempferol treatment using Venn analysis showed 113 common genes (Figure 4). A heat map was generated for all the differentially expressed genes (DEGs) including both groups, using hierarchical clustering (Figure 5).

### 2.5. Ingenuity Pathway Analysis

Pathway analysis of DEGs of the IFN-γ treatment group (295) and IFN-γ + kaempferol treatment group (168) using the ingenuity pathway analysis tool revealed canonical pathways, with their association [−log (*p* value)] and prediction scores (z-score, positive for activation, negative for inhibition, and ND for not predicted). For the IFN-γ + kaempferol treatment group, the most associated, activated, and inhibited pathways were “cardiac hypertrophy signaling, role of NFAT in cardiac hypertrophy, xenobiotic metabolism general signaling pathway, neuroinflammation signaling pathway, role of hypercytokinemia/hyperchemokinemia in the pathogenesis of influenza, interferon signaling, antigen presentation pathway, Th1 and Th2 activation pathway, pyroptosis signaling pathway, role of PKR in interferon induction and antiviral response, B cell development, IL-4 signaling, activation of IRF by cytosolic pattern recognition receptors, MSP–RON signaling in macrophages pathway, and Fc Epsilon RI signaling pathways (Table 3 and Figure 6). Cellular homeostasis, T cell development, autophagy, chemotaxis, organismal death, morbidity or mortality, and viral replicon replication were among the conditions predicted by the ingenuity pathway analysis. Functional enrichment analysis further revealed significantly altered biological processes associated with cholesterol and lipids that might lead to atherosclerosis (Table 4). Inflammation and stress responses (IL4R, FN1, APOE, CSF2RA), cell adhesion and migration (FN1), lipid transport and metabolism (NR1H3, IL4R, RXRA), extracellular matrix molecules (ECM1, IL4R, FN1, RXRA), and transcriptional regulation (NR1H3, RXRA) were all affected by IFN-γ, with or without kaempferol (Table 5 and Figure 7).

### 2.6. SwissTargetPrediction and Protein–Protein Interaction Networks

The SwissTargetPrediction program predicted that oxidoreductase, lyase, nuclear receptor, primary active transporter, and kinase were the bioactive targets of kaempferol and that they may be used to treat atherosclerosis. PPI networks for the IFN-γ + kaempferol group were built using Cytoscape (https://cytoscape.org/ (accessed on 10 April 2022)) and the STRING database (https://string-db.org/ (accessed on 10 April 2022)). We identified 16 nodes in the PPI network: apolipoprotein E (APOE); apolipoprotein C1 (APOC1); CD36 molecule (CD36); hydroxycarboxylic acid receptor 2 (HCAR2); macrophage scavenger receptor 1 (MSR1); nuclear receptor subfamily 1 group H member 3 (N1RH3); retinoid X receptor-alpha (RXRA); scavenger receptor class B member 1. Many of these nodes are related to the regulation of lipid metabolic processes and lipid transport (Figure 8).

## 3. Discussion

Kaempferol is commonly consumed as a phytochemical in a well-balanced diet, as it is enriched in plant-derived foods [21]. It has anti-inflammatory and anti-atherosclerotic effects [22]. To further understand the mechanism of kaempferol activity, we evaluated its influence on monocyte invasion.

IFN-γ is a prospective therapeutic target since it is engaged during the progression of atherosclerotic plaque development [23]. It is the key regulator of atherosclerosis disease progression. IFN-γ may activate two proinflammatory genes, MCP-1 and ICAM-1 [24,25]. Before lipopolysaccharide induction, kaempferol treatment of J774.2 macrophages significantly reduced the amount of MCP-1 mRNA produced [26]. We also discovered that 10 M kaempferol inhibited MCP-1 and ICAM-1 production mediated by IFN-γ. This supported kaempferol anti-inflammatory mechanism of action.

MCP-1 promotes monocyte migration to endothelial cell undersides, whereas ICAM-1 promotes monocyte binding to endothelial cells and migration into the intima [27,28]. It was shown that targeting MCP-1 and ICAM-1 slows the progression of atherosclerosis [29,30,31]. Kaempferol inhibits monocytic cell motility toward MCP-1 and ICAM-1, according to our findings. IFN-γ-treated cells significantly increased macrophage migration, whereas kaempferol administration significantly reduced chemokine-mediated macrophage migration. Our findings are in line with previous research that found kaempferol reduced E-selectin, ICAM-1, VCAM-1, and MCP-1 expression in atherosclerotic aortas [32,33]. As a result, monocytic migration inhibition appears to be a critical mechanism in kaempferol-mediated atheroprotection.

IPA and STRING analysis revealed common genes (SOAT1, SCARB1, APOE, and APOC1) related to lipid metabolism that activate cholesterol effluxes. Gene ontology analysis predicted the following atherosclerosis-related molecular processes: positive regulation of cholesterol storage, very low density lipoprotein particle clearance, cholesterol efflux, phospholipid efflux, low-density lipoprotein particle clearance, reverse cholesterol transport, cholesterol homeostasis, lipid transport, and regulation of hydrolase activity.

In IFN-γ + kaempferol-treated cells, gene expression of pro-atherogenic genes (CXCL9, HLA-DRA, IDO1, CXCL10, IFI44 and GBP5) was reduced compared to IFN-γ-treated cells. Other investigations found that IFN-γ and LPS changed the expression of STAT1-induced genes (CXCL9, CXCL10, and GBP5) in vascular smooth muscle cells in vitro [34]. CXCL10 reduces plaque stability by weakening the fibrous cap, thus increasing the risk of CVD [35,36]. IDO1 was overexpressed in coronary atherosclerotic plaques in activated THP-1 macrophages, which could promote thrombus formation [37]. The lower expression of pro-atherogenic genes can be explained by the upregulation of STAT1 in the IFN-γ treatment group compared to the IFN-γ + kaempferol treatment group.

Many atherosclerosis-related genes, such as ADAMTS2, ECM1, and USP18, showed changes in their expression. According to one study, ADAMTS2 participates in enhancing TGF signaling and is a multilevel regulator of ECM deposition and remodeling [38]. Hardy et al. [39] found high levels of ECM1 expression in aging and infarcted hearts. ECM1 acts as a pro-fibrotic factor by activating ERK1/2 and AKT and stimulating cardiac fibroblasts in vitro [39]. Silencing of USP18 causes an increase in cell-associated ICAM-1, and VCAM-1 and in IL-6 secretion, as well as attenuation of inflammation by blocking the NFB pathway, according to Auclair et al. [40].

Downregulated genes (SYK, SMAD3, S100A4, and FERM3) were expected to be involved in inhibiting cellular recruitment by IPA, and were previously documented to enhance the beginning and progression of atherosclerosis [41]. Furthermore, in the IFN-γ + kaempferol group, deactivation of CD55, COLT1, DIAPH1, DPP7, IL4R, KCNAB2, PRCP, PTPN9, QSOX1, SYK, TSPAN14, and TUBB4b was linked to the suppression of leukocyte, phagocyte, and myeloid cell degranulation. Downregulation of ETV5, IL1R1, and IL4R reduced cellular accumulation, whereas reduction of leukocyte transmigration was induced by the downregulation of TSPAN14 and FREMT3 [42].

The highest-ranking networks related to cardiovascular disease, cellular compromise, and inflammatory response were predicted using IPA. ApoE-HDL is known to utilize reverse cholesterol transport to remove cholesterol from peripheral cells and deliver it to the liver. ApoE and ApoE-HDL decrease VLDL cholesterol and suppress ECM and ECM-1 expression in vascular smooth muscles [43]. By suppressing nuclear factor kappa B (NF-κB) transmission, liver-X-receptors (LXRs) mediate inflammatory responses [44,45]. We discovered that NR1H/LXR and high-density lipoprotein decreased NF-B signaling. These findings suggest that by activating LXRs, kaempferol can counteract proinflammatory effects and lower the possibility of heart problems.

In the IFN-γ + kaempferol group, we discovered a deactivated cardiac hypertrophy pathway. Cardiac hypertrophy raises the risk of atherosclerotic heart disease by altering inflammatory responses and oxidative stress via the ERK cascade (Ras/Raf/MEK/ERK) [46,47,48]. Kaempferol inhibits ERK1/2, which is triggered by the ERK cascade (Ras/Raf/MEK/ERK), by downregulating cytokine receptors. Cardiac failure and hypertrophy are regulated by MAP kinases (MAPKs, (ERK)1/2, JNK, and p38 kinase [49,50]. Kaempferol inhibits the cytochrome P450 1B1 (CYP1B1) and Fc Epsilon RI signaling pathways, according to studies [51,52]; we confirmed this observation. Mast cells stimulated by IgE and FcR1 play a vital role in inflammatory diseases such as atherosclerosis, and the absence of FcR1 reduced inflammation, plaques, and the burden of atherosclerosis [53,54,55]. CYP1B1 is mostly expressed in the liver and plays a role in the metabolism of a wide range of xenobiotics, as well as of arachidonic acid, estrogen, and cholesterol [52,56]. It is important in the etiology of cardiovascular disorders, including heart hypertrophy and hypertension [52]. As a result of Kaempferol potential to suppress the xenobiotics signaling pathway, cardiac hypertrophy signaling and atherosclerosis progression may be inhibited.

## 4. Materials and Methods

### 4.1. Cell Lines

THP-1 cells were grown in RPMI-1640 culture medium (Catalogue No: A1049101) with 10% fetal bovine serum (Catalogue No: A3160802), 1 mM Penicillin–Streptomycin (Catalogue No: 15140122), and 200 mM l-Glutamine (Catalogue No: 25030081) (Thermo Fisher Scientific, Waltham, MA, USA).

### 4.2. Chemicals and Reagents

Sigma-Aldrich provided kaempferol (Catalogue No: 60010), 4-phorbol 12-myristate 13-acetate (PMA) (Catalogue No: J63916), and human interferon (Catalogue No: I17001) (Gillingham, UK). For the preparation of a stock solution, 8 mg of kaempferol (purity ≥97% HPLC) was dissolved in 2.79 mL of DMSO (Catalogue No: D12345) (Invitrogen, Thermo Fisher Scientific, Waltham, MA, USA). For at least 3 months, the stock was stored in the dark at 20 °C. We used Thermo Fisher Scientific’s Pierce LDH cytotoxicity assay kit (Catalogue No: 88954), Qiagen’s RNeasy and QuantiFast SYBR Green PCR Kit (Catalogue No: 74104) (Germantown, MD, USA), Promega’s ImProm-IITM Reverse Transcription System Kit (Catalogue No: A3800) (Madison, WI, USA), and Abcam’s Cholesterol Efflux Assay Kit (Catalogue No: ab196985) (Abcam, Cambridge, UK).

### 4.3. THP-1 Cell Culture

THP-1 cells were generously donated by KFSHRC, Riyadh, Saudi Arabia. T25 or T75 cell culture flask were used in an upright position for sub-culturing THP-1 cells every 48–72 h.

### 4.4. Lactate Dehydrogenase Assay

THP-1 monocytes were plated in 96-well culture plates at a density of 100,000 cells/well, and 160 nM PMA was utilized to differentiate THP-1 in RPMI for the first 24 h in an incubator (5% carbon dioxide *v*/*v*). Different concentrations of kaempferol (5, 10, 25, 50, 75, and 100 µM) were applied to the differentiated cells, which were incubated for another 24 h. Vehicle alone (DMSO treated) was used as a control. At the end of the incubation, positive-control cells were prepared by adding the lysis solution from the LDH kit and then incubating the cells for 45 min. To correct the changes in background absorption, a negative control was also used. After 45 min, 50 μL of cell culture medium was transferred to a sterile 96-well plate and mixed using 50 μL of the mixture supplied in the LDH kit. To stop the reaction, 50 μL of stopping solution was added to each microwell and left at room temperature in the dark for 30 min. The absorption was measured at 490 nm (BioTek Instruments, Winooski, VT, USA). The average value of no-cell control was subtracted from all measured values, and the result was expressed as the percentage of cell survival with respect to control values.

### 4.5. Crystal Violet Assay

After removing the growth medium from the original 96-well plate acquired from the LDH test, the macrophages were subjected to the crystal violet assay. The wells were stained for 5 min at room temperature with 50 μL of a 0.2% (*w*/*v*) crystal violet solution. The macrophages were then washed four times in PBS, and crystal violet was solubilized using 50 μL of a solution of 0.1 M NaH2PO4. The microplate was gently shaken for 5 min and read at 570 nm. Crystal violet, which binds to adhering cells’ DNA, was used to stain the living adherent cells. The amount of living cells vs. that of untreated cells was used to calculate cell viability.

### 4.6. In Vitro Monocyte Migration Assay

The Boyden chamber assay was used to investigate the effects of MCP-1 and ICAM-1 on monocyte migration. Cell inserts containing a filter with 8 μm pores were employed to create two compartments in the wells, so to replicate the artery endothelium layer and allow monocyte movement. IFN-γ-stimulated THP-1 macrophages were then treated with 5 and 10 M kaempferol or vehicle control (0.1% DMSO). Except for the vehicle control, each compartment was filled with the chemokine MCP-1 (20 ng/mL). THP-1 monocytes were added to the top of all cell culture inserts at a density of 5 × 10^5^ cells/well. One ml of complete growth medium was added to the cells. The number of cells that moved through the filter was counted using a hemocytometer to determine monocyte migration.

### 4.7. Microarray Processing and Gene Expression Analysis

Total RNA was extracted by the RNeasy kit (Qiagen, Germantown, MD, USA) from THP-1 cells treated with vehicle control, IFN-γ for 3 h, 5 µM kaempferol in the absence of IFN-γ, and 10 µM f kaempferol in the presence of IFN-γ, following the manufacturer’s instructions. A NanoDrop1000 spectrophotometer (Thermo Fisher Scientific, MA, USA) was used to measure the concentration of RNA. The ImProm-II Reverse Transcription kit was applied to reverse transcribe the RNA template following the manufacturer’s instruction. Affymetrix GeneChips were used to profile transcriptional expression on the Affymetrix platform (Gene 1.0ST, Santa Clara, CA, USA). To process the arrays, target preparation was followed by hybridization, washing, staining, and scanning. Raw data from Affymetrix CEL files were used in Partek Genome Suit 7.0 for image processing and probe quantification. Analysis of variance (ANOVA) was performed to detect the differentially expressed genes using false discovery rate (FDR < 0.05) and fold change (FC > ±2) as cut-off.

### 4.8. Quantitative Real-Time Polymerase Chain Reaction

The SYBR Green Kit from Qiagen was used for qRT–PCR, and cDNA was put into a PCR 96-well plate according to the manufacturer’s procedure. The PCR was carried out using the Applied Biosystems StepOne Plus RealTime PCR System. The target genes were MCP-1 and ICAM-1, while GAPDH was used as a reference. Primer sequences were as follows: MCP-1 forward and reverse- CGCTCAGCCAGATGCAATCAATG and ATGGTCTTGAAGATCACAGCTTCTTTGG; ICAM-1 forward and reverse- ACCAGAGGTTGAACCCCAC and GCGCCGGAAAGCTGTAGAT; GAPDH forward and reverse- CTTTTGCGTCGCCAGCCGAG and GCCCAATACGACCAAATCCGTTGACT. The relative expression of the target genes as compared to that of the reference gene GAPDH and evaluated using the Ct relative gene expression technique once the PCR reaction was completed.

### 4.9. Ingenuity Pathway Analysis

Significant canonical molecular pathways, networks, functions, and were discovered using Ingenuity Pathways Analysis ver 338830M (Ingenuity Systems, Redwood City, CA, USA). IPA (Qiagen, Germantown, MD, USA) was also used for discovering upstream transcriptional regulators.

### 4.10. In Silico Analysis

The machine-readable forms of kaempferol (C15H10O6) were obtained from the PubChem database (Kim, 2016), and SwissTargetPrediction was used to estimate kaempferol’s anti-atherogenic effects. Utilizing a reverse screening technique, ligand-based target prediction was conducted using the similarity principle (Daina and Zoete, 2019). In addition, utilizing the STRING database (https://string-db.org/ (accessed on 4 April 2022)), related biological processes for combined IFN-γ and kaempferol treatment were identified, and a protein interaction network (PPI) was created. The PPI network findings were created using Cytoscape 3.5.0 (http://www.cytoscape.org/ (accessed on 4 April 2022)).

### 4.11. Statistical Analysis

Analytic applications like Excel Microsoft 2010 and GraphPad Prism ver 8 were used to perform one-way ANOVA. After three separate studies, statistical significance was considered for a *p*-value < 0.05.

## 5. Conclusions

Kaempferol modulated monocyte-to-macrophage differentiation and inhibited several atherogenesis processes in THP-1 macrophages, including pro-inflammatory gene expression, monocyte movement, IFN-γ-mediated inflammatory responses, and cholesterol export. Overall, our findings suggest that kaempferol might be utilized to reduce the risk of atherosclerotic illnesses by modifying the expression of genes linked to the disease, perhaps leading to the development of a novel supplementary treatment for cardiovascular disease.

## Figures and Tables

**Figure 1 ijms-23-07461-f001:**
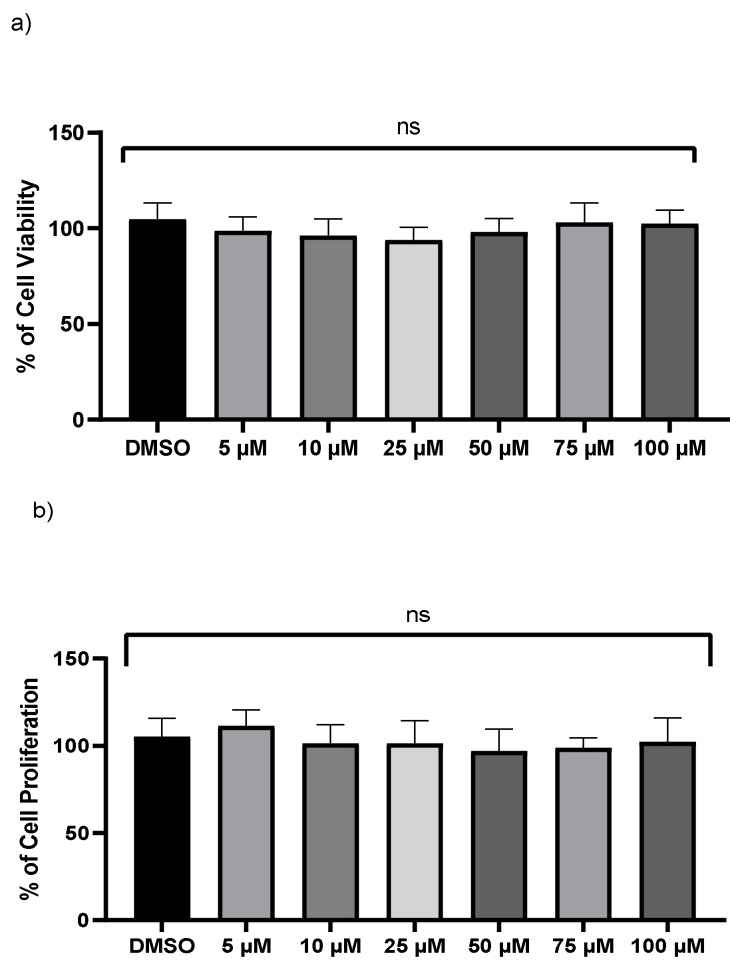
Kaempferol (5–100 µM) does not influence macrophages cell viability. (**a**) Cell viability was assessed by calculating the LDH value in the medium from treated cells, (**b**) the crystal violet test on adherent cells was used to measure cell proliferation; ns—not significant. Mean ± SD of three separate experiments.

**Figure 2 ijms-23-07461-f002:**
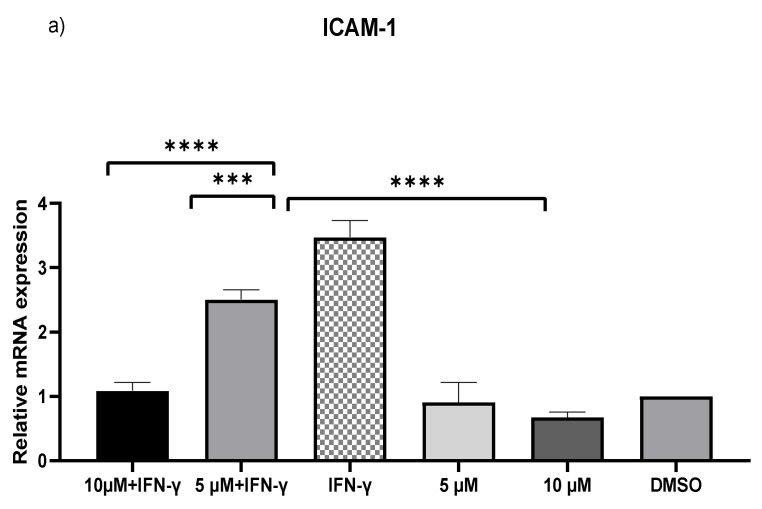
Kaempferol significantly prevents IFN-γ-triggered overexpression of ICAM-1 and MCP-1. To amplify complementary DNA, gene-specific primer sequences were utilized. The mean standard deviation of three separate experiments is reported. Statistical significance was determined by one-way ANOVA, with *** *p* ≤ 0.001, **** *p* ≤ 0.0001.

**Figure 3 ijms-23-07461-f003:**
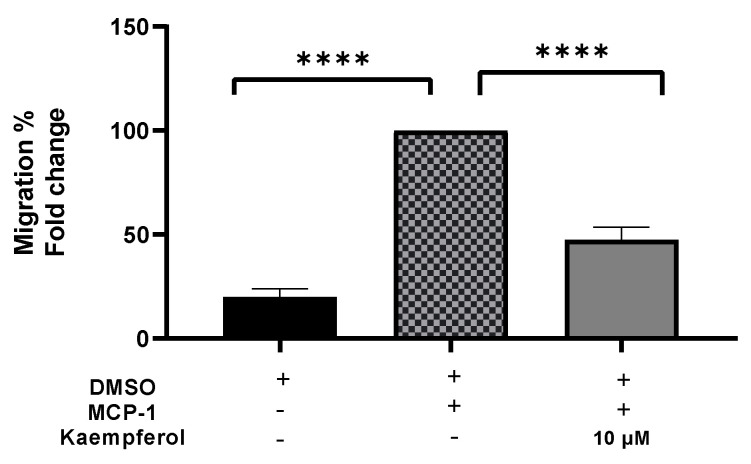
Kaempferol effectively inhibits THP1 monocyte migration triggered by MCP-1. The number of migrated cells was quantified and reported as a proportion of transmigrated cells to measure monocyte migration. The mean standard deviation of three separate experiments is reported. Statistical significance was determined by one-way ANOVA, with **** *p* ≤ 0.0001.

**Figure 4 ijms-23-07461-f004:**
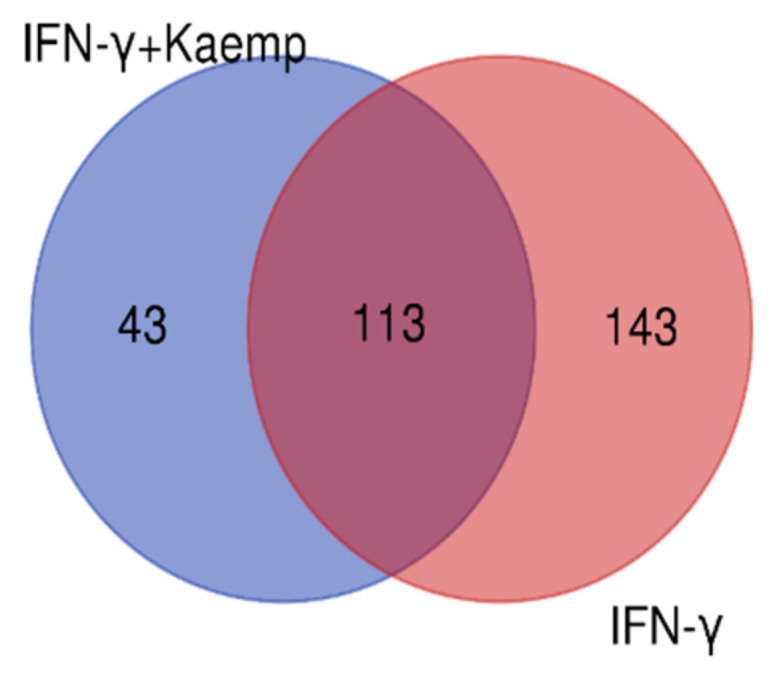
A Venn diagram comparing differentially expressed genes in both groups is shown. Venny 2.1.0 (http://bioinfogp.cnb.csic.es/tools/venny (accessed on 4 April 2022) was used to create the figure.

**Figure 5 ijms-23-07461-f005:**
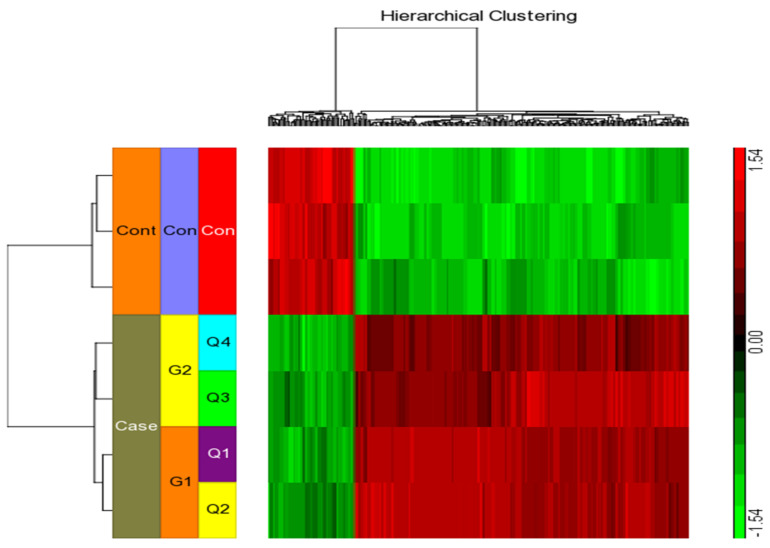
Using the Affymetrix Human ST 1.0 array and the Partek GS 7.0 software, we performed hierarchical clustering and functional analysis on chosen genes that were substantially differently expressed in treated cells.

**Figure 6 ijms-23-07461-f006:**
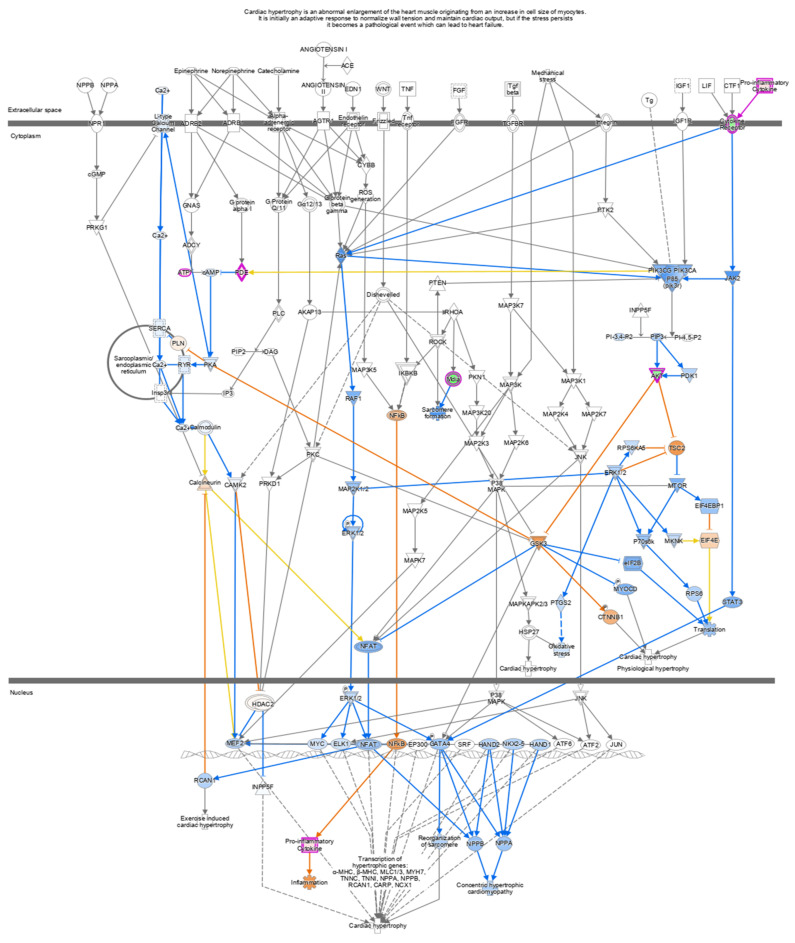
In cells treated with IFN-γ + kaempferol, canonical pathways indicating Cardiac Hypertrophy signaling (enhanced) were found using Ingenuity Pathway Analysis (IPA). Upregulated genes are shown in red, whereas downregulated genes are shown in green.

**Figure 7 ijms-23-07461-f007:**
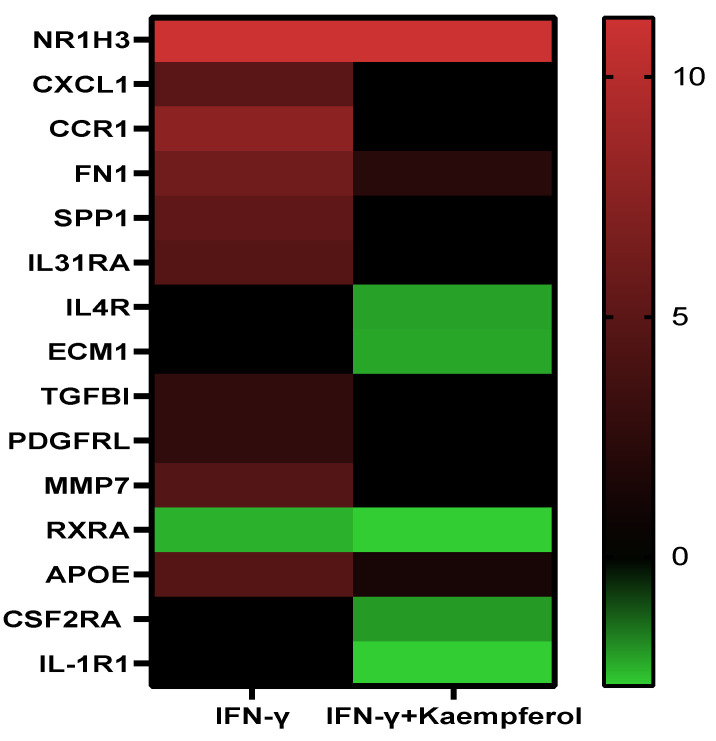
Heatmap displaying the levels of differentially expressed genes in the presence of 130 nM IFN-γ and 130 nM IFN-γ + 10 µM kaempferol. Color code: red for upregulated, green for downregulated genes, and black for non-significant expression difference.

**Figure 8 ijms-23-07461-f008:**
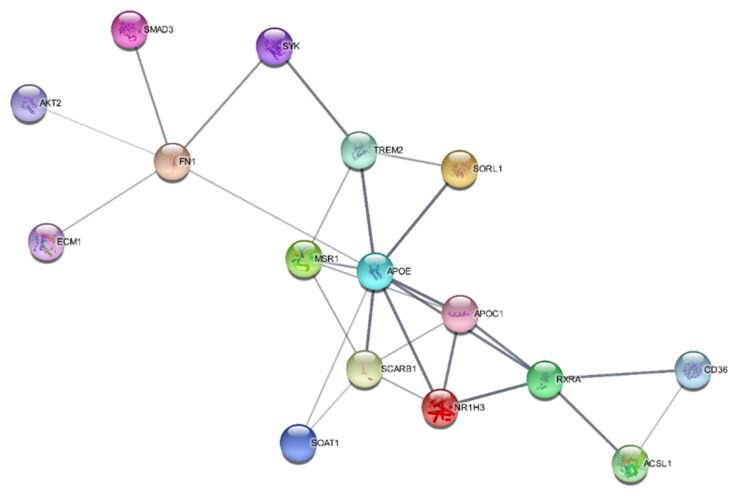
Protein–protein interaction networks constructed for the IFN-γ + kaempferol group, predicted by the STRING database.

**Table 1 ijms-23-07461-t001:** Genes most significantly up- and downregulated by IFN-γ treatment vs. control.

Gene Name	Gene Symbol	*p*-Value	Fold-Change
Upregulated differentially expressed genes			
CXCL9, C-X-C Motif Chemokine Ligand 9	*CXCL9*	2.08 × 10^−6^	197.457
Major Histocompatibility Complex Class II DR Alpha	*HLA-DRA*	1.63 × 10^−5^	137.562
Tryptophan 2,3-Dioxygenase	*TDO2*	0.000551137	133.681
Indoleamine 2,3-Dioxygenase 1	*IDO1*	1.18 × 10^−5^	108.503
C-X-C Motif Chemokine Ligand 10	*CXCL10*	6.52 × 10^−5^	86.0006
Interferon Induced Protein 44 Like	*IFI44L*	0.000593426	70.7593
Serpin Family G Member 1	*SERPING1*	1.73 × 10^−7^	65.1139
Guanylate Binding Protein 5	*GBP5*	1.24 × 10^−6^	59.8055
Downregulated differentially expressed genes			
Sortilin Related Receptor 1	*SORL1*	0.000414781	−8.41001
S100 Calcium Binding Protein A4	*S100A4*	0.000298601	−5.95322
Sugen kinase 223	*SGK223*	0.000583494	−4.32646
SMAD Family Member 3	*SMAD3*	9.74 × 10^−5^	−3.4329
Vasohibin 1	*VASH1*	0.000277198	−3.30646
Long Intergenic Non-Protein Coding RNA 1001	*LINC01001*	8.77 × 10^−5^	−3.21825
Filamin B	*FLNB*	2.46 × 10^−5^	−3.1648
Proteolipid Protein 2	*PLP2*	0.000246883	−3.0025
glutathione reductase	*GSR*	0.000199734	−2.9929
Neurotensin Receptor 1	*NTSR1*	6.73 × 10^−5^	−2.85795

**Table 2 ijms-23-07461-t002:** Genes most significantly differentially up- and downregulated by IFN-γ + kaempferol treatment vs. control).

Gene Name	Gene Symbol	*p*-Value	Fold-Change
Upregulated differentially expressed genes			
C-X-C Motif Chemokine Ligand 9	*CXCL9*	2.35 × 10^−6^	167.766
Major Histocompatibility Complex, Class II, DR Alpha	*HLA-DRA*	2.06 × 10^−5^	103.719
Tryptophan 2,3-Dioxygenase	*AIM2/TDO2*	2.71 × 10^−6^	66.5112
Indoleamine 2,3-Dioxygenase 1	*IDO1*	1.91 × 10^−5^	63.8001
C-X-C Motif Chemokine Ligand 10	*CXCL10*	9.54 × 10^−5^	57.1592
Interferon Induced Protein 44 Like	*IFI44*	1.59 × 10^−6^	47.0317
Guanylate Binding Protein 5	*GBP5*	1.59 × 10^−6^	46.794
Serpin Family G Member 1	*SERPING1*	2.81 × 10^−7^	40.4335
Purinergic Receptor P2X 7	*P2RX7*	1.25 × 10^−5^	38.8665
Guanylate Binding Protein 1	*GBP1*	2.67 × 10^−5^	36.667
Downregulated differentially expressed genes			
Sortilin Related Receptor 1	*SORL1*	0.0004207	−8.34514
S100 Calcium Binding Protein A4	*S100A4*	0.000280278	−6.12928
Proteolipid Protein 2	*PLP2*	4.19 × 10^−5^	−5.60529
NAD(P)H Quinone Dehydrogenase 1	*NQO1*	0.000382832	−5.44158
Uronyl-2-sulfotransferase	*UST*	0.000405202	−5.24763
CD37 molecule	*CD37*	3.09 × 10^−5^	−5.01109
xylosyltransferase I	*XYLT1*	0.000445907	−4.90686
Rho GDP dissociation inhibitor (GDI) beta	*ARHGDIB*	0.000154905	−4.55132
Quiescin Q6 sulfhydryl oxidase 1	*QSOX1*	5.81 × 10^−5^	−3.92237
Cytochrome b561 family, member A3	*CYB561A3*	6.47 × 10^−5^	−3.68936

**Table 3 ijms-23-07461-t003:** Most associated, activated, and inhibited canonical pathways canonical pathways identified in the “IFN-γ + kaempferol” treatment group.

Ingenuity Canonical Pathways	−log (*p*-Value)	z-Score	Molecules
Cardiac Hypertrophy Signaling (Enhanced)	12.19	−1.225	ACVR2A, ADCY8, ADCY9, AKT2, CALM1, CAMK2G, DIAPH1, DIAPH2, GNA13, HSPB1, IKBKE, IL17RA, IL1R1, IL31RA, ITGA2, ITGB3, JAK2, MKNK1, MRAS, PLCG2, PNPLA8, RCAN1, SMPDL3A, SRF, TGFBR1, TGFBR2, TNFSF10, TNFSF13B, TSC2
Role of NFAT in Cardiac Hypertrophy	1.995	−2.111	ADCY8, ADCY9, AKT2, CALM1, CAMK1, CAMK2G, MRAS, PLCG2, RCAN1, TGFBR1, TGFBR2
Cardiac Hypertrophy Signaling	1.893	−1.541	ADCY8, ADCY9, CALM1, GNA13, HSPB1, MRAS, MYL6, PLCG2, RND3, SRF, TGFBR1, TGFBR2
Xenobiotic Metabolism General Signaling Pathway	4.813	−1.541	
Role of Hypercytokinemia/hyperchemokinemia in the Pathogenesis of Influenza	12.8	3.464	CXCL10, DDX58, EIF2AK2, IFIT2, IFIT3, ISG20, MX1, OAS1, OAS3, RSAD2, STAT1, STAT2
Interferon Signaling	10.5	2.828	IFI35, IFIT3, IFITM1, IRF1, MX1, OAS1, STAT1, STAT2
Antigen Presentation Pathway	10.2	NP	B2M, CD74, HLA-DPB1, HLA-DQA1, HLA-DQA2, HLA-DQB1, HLA-DRA, TAP2
Th1 and Th2 Activation Pathway	7.04	NP	CD274, HLA-DPB1, HLA-DQA1, HLA-DQA2, HLA-DQB1, HLA-DRA, IL4R, IRF1, mir-29, STAT1
Fc Epsilon RI Signaling	6.41	1.71	AKT2, INPP5D, LCP2, LYN, MRAS, PLCG2, SYK, VAV3
Pyroptosis Signaling Pathway	6.95	2.828	AIM2, GBP1, GBP2, GBP3, GBP4, GBP5, GBP7, P2RX7
Role of PKR in Interferon Induction and Antiviral Response	3.76	2.449	DDX58, EIF2AK2, IFIH1, IRF1, STAT1, STAT2
Activation of IRF by Cytosolic Pattern Recognition Receptors	4.36	2.236	DDX58, IFIH1, IFIT2, STAT1, STAT2
Necroptosis Signaling Pathway	3.38	2.236	ATP, AXL, EIF2AK2, STAT1, STAT2, TNFSF10
MSP-RON Signaling In Macrophages Pathway	4.08	−2.449	HLA-DPB1, HLA-DQA1, HLA-DQA2, HLA-DQB1, HLA-DRA, STAT1
Neuroinflammation Signaling Pathway	6.14	1.897	AKT2, ATP, B2M, CXCL10, HLA-DPB1, HLA-DQA1, HLA-DQA2, HLA-DQB1, HLA-DRA, P2RX7, STAT1, SYK
IL-4 Signaling	5.79	NP	AKT2, HLA-DPB1, HLA-DQA1, HLA-DQA2, HLA-DQB1, HLA-DRA, IL4R
B Cell Development	5.2	NP	HLA-DPB1, HLA-DQA1, HLA-DQA2, HLA-DQB1, HLA-DRA

**Table 4 ijms-23-07461-t004:** Biological processes derived from functional enrichment networks associated with genes affected by IFN-γ + kaempferol treatement.

Biological Process	Strength	*p* Value	Molecules
Positive regulation of cholesterol storage	2.72	2.29 × 10^−5^	MSR1, SCARB1, CD36
Very-low-density lipoprotein particle clearance	2.61	0.0021	APOE, APOC
Cholesterol efflux	2.33	4.75 × 10^−5^	SOAT1, SCARB1, APOE, APOC1
Phospholipid efflux	2.31	0.0049	APOE, APOC1
Low density lipoprotein particle clearance	2.18	0.00028	SOAT1, SCARB1, CD36
Reverse cholesterol transport	2.16	0.0075	SCARB1, APOE
Cholesterol homeostasis	1.76	0.00016	SOAT1, SCARB1, APOE, NR1H3
Lipid transport	1.52	8.04 × 10^−8^	SOAT1, MSR1, APOE, SCARB1, APOC1, RXRA, CD36, ACSL1
Regulation of hydrolase activity	0.88	0.0050	SORL1, SYK, FN1, ECM1, AKT2, SMAD3, N1RH3, APOC1

**Table 5 ijms-23-07461-t005:** Effect of IFN-γ and IFN-γ + kaempferol treatments on the mRNA expression of important genes involved in atherosclerosis.

Gene Function	IFN-γ	IFN-γ + Kaempferol
Stress response	CCR1↑ SPP1↑ CXCl1↑ FN1↑ APOE↑	IL4R↓ FN1↑ APOE↑ CSF2RA ↓ IL-1R1↓
Apoptosis	TGFBI↑ SPP1↑	
Blood coagulation and circulation	PDGFRL↑	
Cell adhesion	SPP1↑ FN1↑	FN1↑
Extracellular Matrix Molecules	MMP7 ↑ FN1↑	ECM1↓ IL4R↓ FN1↑ RXRA↓
Lipid transport and metabolism	RXRA↓ NR1H3↑	IL4R↓ NR1H3↑ RXRA↓
Cell growth and proliferation	PDGFRL↑ SPP1↑ IL31RA↑ TGFBI↑	IL4R↓
Transcription regulation	RXRA↓ NR1H3↑	RXRA↓ NR1H3↑

Note: ↑ represents induction, and ↓ represents reduction in expression.

## Data Availability

The raw datasets used in this study are available at NCBI’s GEO repository with accession number of GSE160430.

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
