# Peer review of "Investigation of the Molecular Mechanisms Underlying the Antiatherogenic Actions of Kaempferol in Human THP-1 Macrophages"

_ijms, 2022, doi:10.3390/ijms23137461_

Round 1

Reviewer 1 Report

Article concerns about kaempferol as an effective therapy for atherosclerosis. A very large panel of protein expression was analyzed in cells under the influence of kaempferol.

I miss information on how the kaempferol concentrations used were selected "Different concentrations of kaempferol (5, 10, 25, 50, 75, and 100 M) were applied to differentiated cells" Here "5 μM of kaempferol in the absence of IFN-γ and 10 μM of kaempferol in the presence of IFN-γ" concentration of kaempferol is 1000 times smaller, why?

Editorial comments :

Figure 1, 2a, 3 and 4 can be a little smaller

In the description of figure 7 "10 m kaempferol" should be corrected

Proofreading of references is required. Eg there is no publication title in reference 1. In some references there are underlines of surnames

Author Response

Article concerns about kaempferol as an effective therapy for atherosclerosis. A very large panel of protein expression was analyzed in cells under the influence of kaempferol.

Response: Thanks for review and suggestions

I miss information on how the kaempferol concentrations used were selected "Different concentrations of kaempferol (5, 10, 25, 50, 75, and 100 μM) were applied to differentiated cells" Here "5 μM of kaempferol in the absence of IFN-γ and 10 μM of kaempferol in the presence of IFN-γ" concentration of kaempferol is 1000 times smaller, why?

Response: Before showing kaempferol specific therapeutic potential, it was mandatory to meks sure that it does not have any cytotoxicity effect on normal cells like monocyte-derived THP-1 macrophages. Therefore different concentration of kaempferol were used for crystal violet test to examine cell proliferation and LDH assay to examine cell viability.

It was typo error and corrected. Kaempferol treating concentrations were 5-100 μM.

Editorial comments:

Figure 1, 2a, 3 and 4 can be a little smaller

Response: Figure’s size were reduced and could further be adjusted by production team (if needed).

In the description of figure 7 "10 m kaempferol" should be corrected

Response: Corrected to 10 µM kaempferol

Proofreading of references is required. Eg there is no publication title in reference 1. In some references there are underlines of surnames

Response: Reference 1 is complete. It has one word title “Atherosclerosis”. Please find PubMed and DOI link:

Pubmed Link: https://pubmed.ncbi.nlm.nih.gov/11001066/ and

DOI: https://www.nature.com/articles/35025203

Underlines of surnames and hyperlink have been removed from references.

Reviewer 2 Report

Dear Authors

This is an original article based on the in vitro evaluation of kaempferol  models for inflammation processes in pain and other relater diseases.

The topic is within the aim and scope of the journal and the text well organized. Please add statystical analysis as a separate paragraph for each diverse assay.

When validating your compound, have you considered the influence of other co-factors involved in inflammation?.

Recently a lot of works have been published on the influence of natural compounds on CVD-inflammatory based process, it could be useful to improve the manuscript introduction considering such background. Find below some literature: "An overview on plants cannabinoids endorsed with cardiovascular effects", "Calceolarioside A, a Phenylpropanoid Glycoside from Calceolaria spp., Displays Antinociceptive and Anti-Inflammatory Properties".

Author Response

This is an original article based on the in vitro evaluation of kaempferol models for inflammation processes in pain and other related diseases.

Response: Thanks for through review and suggestions. We did English language editing to improve it.

The topic is within the aim and scope of the journal and the text well organized. Please add statistical analysis as a separate paragraph for each diverse assay.

Response: Page 15 (heading 4.11), statistical analysis method was written as “The analytic applications like Excel Microsoft 2010 and GraphPad Prism ver 8 were used to do the one-way ANOVA. After three separate studies, statistical significance was considered for p-value <0.05”. Writing it again for each diverse assay won’t add value to manuscript and appear as a repetition.

When validating your compound, have you considered the influence of other co-factors involved in inflammation?

Response: No. However, your suggestion will be implemented in future studies.

Recently a lot of works have been published on the influence of natural compounds on CVD-inflammatory based process, it could be useful to improve the manuscript introduction considering such background. Find below some literature: "An overview on plants cannabinoids endorsed with cardiovascular effects", "Calceolarioside A, a Phenylpropanoid Glycoside from Calceolaria spp., Displays Antinociceptive and Anti-Inflammatory Properties".

Response: We gone through suggested manuscript and included following references in present manuscript.

  • Pieretti, S.; Saviano, A.; Mollica, A.; Stefanucci, A.; Aloisi, A.M.; Nicoletti, M. Calceolarioside A, a Phenylpropanoid Glycoside from Calceolaria spp., Displays Antinociceptive and Anti-Inflammatory Properties. Molecules 2022, 27, 2183.
  • Dimmito MP, Stefanucci A, Della Valle A, Scioli G, Cichelli A, Mollica A. An overview on plants cannabinoids endorsed with cardiovascular effects. Biomed Pharmacother. 2021 Oct;142:111963.